# XFeat Revisited: Reproducibility and Evaluation of a Lightweight Image Matcher

## Abstract

We present a reproducibility study of XFeat, a lightweight local feature extractor and matcher designed to identify corresponding points across images efficiently on resource-constrained hardware. We re-implement the architecture based on the paper and supplementary material, re-evaluate the authors' released checkpoint alongside our re-implementation, and conduct additional architectural ablations to examine design choices that were not fully justified in the original work. This distinction between re-evaluation and reproduction is important, as the paper, supplement, and public code differ in several implementation details, including the backbone layout, fusion block, and training losses. Empirically, our reproduced models closely match and, in some cases, outperform the re-evaluated original checkpoint on MegaDepth-1500 and ScanNet-1500, supporting the main claim that XFeat provides a strong accuracy–efficiency trade-off for standard image-matching benchmarks. Our ablations provide a more nuanced view of two architectural arguments from the original paper. In particular, the parallel keypoint branch is important for semi-dense matching, but its benefit is less pronounced than originally claimed, while the evidence for the specific placement of the single skip-connection remains inconclusive. Finally, we reproduce the original downstream evaluations and find close agreement for homography estimation, while Aachen visual localization remains below the reported results, even for the released checkpoint, suggesting sensitivity to underspecified evaluation details. We then extend the analysis to zero-shot out-of-distribution and cross-modal matching across retinal, thermal–visible, and multimodal remote-sensing imagery, where XFeat remains effective in some settings but degrades sharply under severe modality shifts.

## 1 Introduction

Many computer vision tasks require finding the same physical points across two or more images of the same scene. This correspondence problem underlies structure-from-motion, which reconstructs a scene in 3D (Schonberger & Frahm, 2016); visual localization, which estimates a camera's position and orientation (Sattler et al., 2018); SLAM, which jointly tracks a camera and maps its surroundings (Mur-Artal et al., 2015); and homography estimation, which aligns different views of a planar scene (Hartley & Zisserman, 2004). Once matching points are found across images, the underlying geometry can be recovered from their relative positions. A common approach is to detect a sparse set of distinctive image points (keypoints) and describe each one with a compact numerical signature (a descriptor), allowing locations in different images to be matched by comparing their descriptors. Recent learned methods have made correspondence estimation more reliable, but often require larger backbone networks, higher-dimensional descriptors, or additional processing stages (Zhao et al., 2022). Efficient alternatives therefore remain important for applications such as robotics and augmented reality, where correspondence must often be estimated in real time on devices with limited memory and computational capacity (Mur-Artal & Tardós, 2017).

XFeat directly targets this trade-off between matching accuracy and computational cost. In *XFeat: Accelerated Features for Lightweight Image Matching* (Potje et al., 2024), the authors introduce a compact convolutional network for efficient local feature extraction and matching. XFeat is designed to detect keypoints and compute descriptors at practical speeds, even on a CPU, while maintaining competitive matching

accuracy compared to more computationally demanding methods. The architecture supports two modes of operation, a sparse mode, XFeat, which retains a predefined number of the most confident keypoints, and a semi-dense mode, XFeat*, which first considers a much larger set of approximate correspondences and then refines their locations. This produces more matches and can improve geometric estimation at a moderately higher computational cost. The two modes allow users to choose between faster sparse matching and denser correspondence estimates when a downstream task benefits from additional geometric constraints.

These claims make XFeat a useful case study for testing whether an efficient vision model can be reproduced from its paper and whether its performance holds across different implementations, tasks, and image domains.

### 1.1 XFeat Background

XFeat processes each image through two lightweight parallel pathways. At a high level, the model must determine where potentially repeatable points are located, how each location should be represented for comparison, and which candidate representations are reliable enough to match. To keep computation low, XFeat uses very few channels while keeping feature maps large, increasing channel capacity only after spatial downsampling, where convolution is less expensive. A convolutional backbone extracts multi-scale features that are combined into a dense descriptor map, while a reliability head estimates how useful each descriptor is likely to be for matching. Separately, the keypoint branch operates directly on the input image and predicts precise, pixel-level locations of distinctive points. The reliability scores help select useful image locations, while descriptor similarity is used to establish correspondences between them.

Two design choices are of particular interest to our reproducibility study. First, XFeat separates keypoint detection from descriptor extraction. Rather than predicting keypoints from the descriptor backbone, it uses a dedicated parallel branch that operates on low-level image information. The original paper argues that this separation is especially important in the semi-dense mode, where initially coarse matches must be further refined. Second, the backbone includes a single skip-connection that passes early image features directly to a later layer. Its purpose is to preserve fine spatial details that might otherwise be lost as the feature maps are downsampled. We later test both design choices by comparing the default architecture with a coupled keypoint detector and several alternative skip-connection designs.

The paper, supplementary material, and released code describe most of the model, but they disagree on several implementation details. For example, the backbone layout and fusion block described in the paper and supplement differ from the released `model.py`, while the implemented training losses do not fully match those presented in the paper. These differences make the intended implementation ambiguous and highlight the importance of identifying which conclusions remain valid with the exact implementation described in the paper.

Our study makes three contributions. First, we provide a re-implementation guided by the paper and supplementary material, using the official code only when the paper omits implementation-critical details. Second, we re-evaluate the released checkpoint and reproduce the original matching, homography estimation, and visual localization experiments. We further test zero-shot out-of-distribution and cross-modal transfer on retinal, thermal–visible, and multimodal remote-sensing imagery. Third, we test whether XFeat benefits from its specific skip-connection and revisit the question of whether keypoint detection should remain separate from descriptor extraction. Together, these experiments support XFeat's main accuracy–efficiency claim, but provide weaker evidence for some architectural explanations and reveal clear limitations under severe modality shifts. All code is available on GitHub at `github.com/ML-anonymous-researcher/xfeat-reproduction`.

## 2 Scope of reproducibility

The central claim of the original paper is that a lightweight convolutional network can achieve a strong trade-off between matching accuracy and computational cost, without relying on hardware-specific optimizations. The paper attributes this result to three main components: a compact backbone for extracting descriptors, a separate branch for detecting keypoints, and a lightweight module that refines coarse matches in the semi-dense mode. Our goal is to reproduce the main empirical findings, revisit architectural choices that were

only briefly evaluated, and test alternatives that were not considered in the original paper. In particular, we evaluate the following claims:

- **Claim 1:** XFeat provides competitive matching accuracy at practical CPU inference speeds without relying on hardware-specific optimizations.

- **Claim 2:** Separating keypoint detection from descriptor extraction improves semi-dense matching performance.

- **Claim 3:** XFeat benefits from a single skip-connection, and this design is preferable to alternative skip-connection designs.

- **Claim 4:** XFeat provides competitive results in downstream homography estimation and visual localization.

Beyond these original claims, we extend the evaluation to examine how well XFeat transfers without additional training to out-of-distribution and cross-modal imagery, including retinal, thermal–visible, and multi-modal remote-sensing data.

Our study goes beyond reproducing the main results tables by examining design choices that were not fully justified in the original paper. Because discrepancies can arise from re-implementation, the evaluation pipeline, or missing implementation details, we, whenever possible, compare the reported results with our re-evaluation of the released checkpoint and our independently trained model. This helps distinguish failures of model reproduction from differences in evaluation.

## 3 Methodology

Our reproduction is based on the main XFeat paper, its supplementary material, and the official GitHub repository. We treat the paper and supplement as the primary specification of the method and use the released code to resolve implementation details not described in either source. When the sources disagree, we adopt the choice we consider most faithful to the intended method. We refer to the resulting model as our reproduction. All discrepancies and the choices we made are summarized in Section 3.6. This setup reduces the number of arbitrary decisions while keeping the reproduced model as close as possible to the published method. Throughout the paper, *Original* denotes values reported by the XFeat authors, *Re-evaluated* denotes our evaluation of their released checkpoint, and *Reproduced* denotes our independently trained paper-guided model.

For training, we follow the setup described in the paper and repository. The training data combines MegaDepth (Li & Snavely, 2018) with a 20,000-image subset of COCO 2017 (Lin et al., 2014) in a 6:4 ratio. MegaDepth contains Internet photo collections with reconstructed camera geometry and depth, while COCO contains real-world indoor and outdoor scenes from which synthetic image pairs are generated through geometric warping. All training images are resized to $800 \times 600$.

We evaluate XFeat using the same main benchmarks as the original paper. Relative pose estimation is measured on MegaDepth-1500 and ScanNet-1500 (Sarlin et al., 2020). MegaDepth-1500 contains outdoor image pairs sampled from MegaDepth, while ScanNet-1500 is derived from ScanNet (Dai et al., 2017), an RGB-D dataset of reconstructed indoor environments. We also reproduce the original downstream evaluations on HPatches (Balntas et al., 2017), which contains planar image sequences with viewpoint and illumination changes for homography estimation, and Aachen Day-Night 1.0 (Sattler et al., 2018), which evaluates visual localization under severe day-night illumination changes.

Beyond reproducing the benchmarks used in the original paper, we extend the evaluation in two directions. First, we test geometric robustness on RUBIK (Loiseau & Bourmaud, 2025), a camera-pose benchmark containing 16.5k image pairs from nuScenes (Caesar et al., 2020), organized into 33 difficulty levels based on scene overlap, scale ratio, and viewpoint change. Second, we evaluate zero-shot out-of-distribution and cross-modal transfer on FIRE (Hernández-Matas et al., 2017), a retinal registration dataset with 134 manually

annotated image pairs, and the official test split of MTV (Liu et al., 2022b), a multi-view thermal–visible aerial dataset containing five scenes with camera-pose supervision. We further evaluate 600 image pairs from three remote-sensing subsets of the multimodal dataset released with SRIF (Li et al., 2023), covering optical–optical, optical–infrared, and optical–SAR registration.

The official XFeat repository provides end-to-end training instructions and evaluation scripts for MegaDepth-1500 and ScanNet-1500, and explicitly notes that small AUC deviations can result from using RANSAC (Fischler & Bolles, 1981). However, it does not provide complete evaluation pipelines for all experiments, particularly for HPatches and Aachen. We therefore implemented our own scripts for CPU inference speed timing, homography estimation, and visual localization, following the corresponding benchmark protocols as closely as possible. For RUBIK, we use the evaluation code provided by the official repository and for FIRE, MTV, and SRIF, we implement our own evaluation scripts.

## 3.1 Model

Building on Section 1.1, our reproduced XFeat model combines the paper, the supplementary material, and the released code, resolving disagreements between them as described in Section 3.6. Both the keypoint detector and descriptor backbone receive an RGB image $\mathbf{I} \in \mathbb{R}^{H \times W \times 3}$, which is first converted to grayscale. The main building unit of XFeat is the *basic layer*, which consists of a $3 \times 3$ convolution followed by BatchNorm and ReLU. The backbone contains six convolutional blocks. The first two contain two sequential basic layers each, while the remaining four contain three basic layers each. The first basic layer of each block adjusts the channel dimension to $\{4, 8, 24, 64, 64, 128\}$, respectively, while the remaining basic layers preserve the input channel dimension. The first block also preserves the input resolution, while the first layer of each subsequent block uses stride 2 convolution. The single skip-connection average-pools the input grayscale image to $1/4$ resolution, projects it to 24 channels using a $1 \times 1$ convolution, and adds it to the output of the third block.

The descriptor branch combines the outputs of the final three backbone blocks, corresponding to resolutions $\{1/8, 1/16, 1/32\}$. The $1/16$ and $1/32$ feature maps are bilinearly upsampled to $1/8$ resolution, while the latter is also projected from 128 to 64 channels via a $1 \times 1$ convolution. Their sum is processed by a fusion block consisting of two basic layers followed by a $1 \times 1$ convolution, producing a dense descriptor map $\mathbf{F} \in \mathbb{R}^{H/8 \times W/8 \times 64}$. A separate reliability head, consisting of two basic layers followed by a $1 \times 1$ convolution and sigmoid activation, assigns a confidence score to each descriptor, producing $\mathbf{R} \in \mathbb{R}^{H/8 \times W/8}$.

Keypoint detection is performed by a parallel branch that operates directly on the grayscale input rather than the backbone features. The image is divided into non-overlapping $8 \times 8$ cells, whose 64 pixel values are rearranged into channels and processed by three basic layers. A final $1 \times 1$ convolution predicts a keypoint tensor $\mathbf{K} \in \mathbb{R}^{H/8 \times W/8 \times 65}$, containing 65 logits per cell, one for each of its 64 pixel positions and one additional *dustbin* class indicating that the cell contains no keypoint. During inference, the dustbin channel is discarded, and the remaining responses are rearranged into a full-resolution keypoint heatmap. Figure 1 provides a complete overview of both detector and descriptor branches.

In sparse matching, the 65-channel keypoint logits are converted into a full-resolution detection heatmap, then non-maximum suppression is applied to identify candidate keypoints. Each candidate is ranked using the product of its detection score and the corresponding reliability score, and only the highest-scoring keypoints are retained. Their 64-dimensional descriptors are interpolated from the dense descriptor map and matched across the two images using mutual nearest-neighbor search based on cosine similarity. In semi-dense matching, a separate MLP refines each coarse correspondence. It concatenates two 64-dimensional descriptors, processes the resulting 128-dimensional vector through four fully connected layers with 512 units each, followed by BatchNorm and ReLU, and outputs 64 logits corresponding to the possible offsets within an $8 \times 8$ grid.

## 3.2 Training objective

Training XFeat means supervising each of the four outputs described in the previous section: the dense descriptor map, the reliability map, the keypoint logits, and the fine-matching offsets. The paper defines

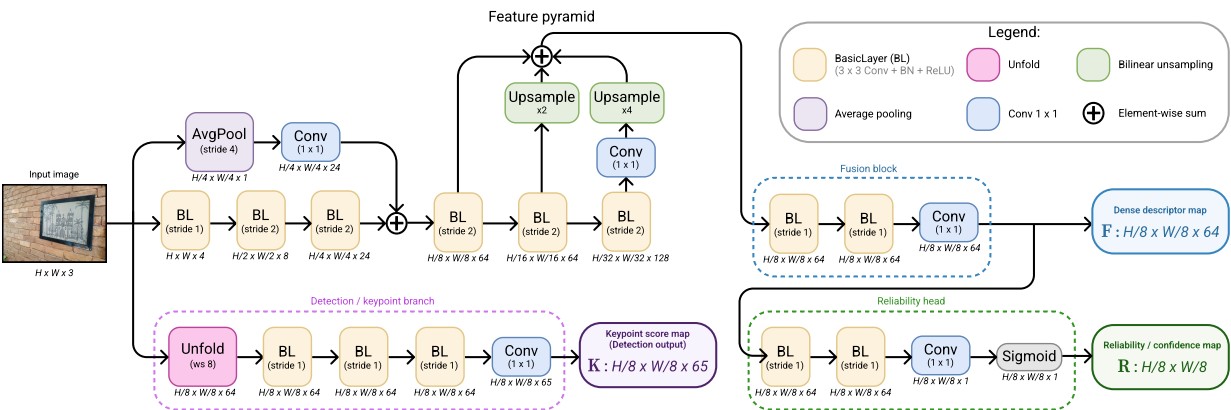

Figure 1: Reproduced XFeat architecture. A six-block convolutional backbone with a skip-connection extracts features at multiple scales, forming a feature pyramid. A fusion block combines these features to generate keypoint descriptors and a reliability map for matching confidence. A separate keypoint head predicts likely keypoint locations within $8 \times 8$ pixel cells.

one loss for each output and combines them into a single weighted sum,

$$\mathcal{L} = \alpha\mathcal{L}_{ds} + \beta\mathcal{L}_{rel} + \gamma\mathcal{L}_{fine} + \delta\mathcal{L}_{kp}.$$

$\mathcal{L}_{ds}$ is a dual-softmax descriptor loss, i.e., for each image pair, it pulls every keypoint's descriptor closer to its true correspondence than to any other descriptor, checked in both directions (image A to B and B to A). $\mathcal{L}_{rel}$ supervises the reliability head using the confidence scores that this same dual-softmax matching produces, so the model learns to predict high reliability for descriptors that are correctly matched and low reliability for those that are not. $\mathcal{L}_{fine}$ supervises the semi-dense refinement step. The paper frames this as a classification problem, in which the model must identify the correct one of 64 possible offsets within each $8 \times 8$ cell. $\mathcal{L}_{kp}$ trains the keypoint branch by distillation. Instead of hand-labeled keypoints, the model is trained to reproduce the keypoints predicted by a separate, already-trained detector, ALIKE (Zhao et al., 2022).

The released training code implements this differently from the formula above. It keeps the same dual-softmax descriptor loss, but replaces $\mathcal{L}_{fine}$ with a different formulation: a confidence-weighted classification loss over coordinates, rather than the cell-classification loss described in the paper. It also splits the remaining supervision differently. Rather than a single reliability term $\mathcal{L}_{rel}$, the code adds a separate L1 loss that regresses the reliability heatmap directly toward the dual-softmax confidence values and applies the ALIKE distillation loss to the 65-way keypoint logits. All terms are then combined by simple averaging, rather than with the explicit weights $\alpha, \beta, \gamma, \delta$ in the paper's formula. In short, the paper and the code agree on *what* needs to be supervised, but disagree on *how* two of the four terms are computed.

Since this deviation could affect reproduction, our default model follows the paper's loss formulation wherever it is fully specified, and falls back to the released code only for details the paper omits entirely. We deliberately do not tune the loss weights $\alpha, \beta, \gamma, \delta$ because the paper does not report specific values, and we do not wish to introduce our own hyperparameter search. Section 3.4 tests the effect of the code's loss formulation directly.

### 3.3 Architectural ablation design

Our architectural ablation study examines two design choices specific to XFeat: the separation of keypoint detection from descriptor extraction and the use of a single input-to-backbone skip-connection. First, *XFeat Coupled Detector* removes the independent image-based detector and instead predicts keypoints from the shared backbone features. This corresponds to our re-implementation of the paper's *joint keypoint extraction* experiment and tests whether a separate detector branch is beneficial. We then evaluate three skip-connection

variants. *XFeat No Skip* removes the single original skip-connection, *XFeat Input Skips* adds appropriately downsampled input projections before every backbone block, and *XFeat ResNet Skips* adds residual connections around each block following the ResNet design (He et al., 2016). Together, these variants test whether the original single skip-connection is necessary or whether alternative skip-connection designs are more effective.

### 3.4 Loss ablation design

To directly measure whether the loss discrepancy described in Section 3.2 affects reproduction, we train one additional ablation variant, *XFeat Code Loss*, which uses the default architecture from Section 3.1 but replaces the paper's loss formulation with the one implemented in the released code, i.e., the confidence-weighted coordinate classification loss in place of $\mathcal{L}_{fine}$, separate reliability and keypoint terms in place of a single weighted $\mathcal{L}_{rel}$ and $\mathcal{L}_{kp}$, and simple averaging in place of the paper's explicit weights. This variant is trained and evaluated identically to *XFeat Default*, isolating the effects of the loss discrepancy from the architectural choices tested in Section 3.3.

### 3.5 Implementation details

Unless stated otherwise, sparse XFeat extracts up to 4,096 keypoints ranked by the product of keypoint confidence and descriptor reliability. XFeat* extracts dense features from each image at two scales, using copies resized to 60% and 130% of the original dimensions. It retains the 10,000 most reliable features across both scales, maps their locations back to the original image coordinates, and refines the resulting coarse matches to obtain more precise correspondences.

The supplement reports training with batches of 10 image pairs using Adam, an initial learning rate of $3 \times 10^{-4}$, and a learning-rate decay of 0.5 every 30,000 iterations. Training converges after 160,000 iterations, taking approximately 36 hours on a single RTX 4090 and using about 6.5 GB of GPU memory. We train our reproduced models on a Slurm cluster using an NVIDIA H100 80 GB GPU, where training takes approximately 17 hours per model. Evaluation is performed on an NVIDIA L4 24 GB GPU. CPU timing experiments are conducted on an Apple M1 Pro and an AMD Ryzen 7 5800H.

Following the original paper, we report inference speed in frames per second (FPS) as the mean over 30 images ± standard deviation at VGA resolution (640 × 480 pixels). Each architecture was trained once. Therefore, the reported uncertainty reflects variation across evaluation pairs rather than variability across independent training runs. For all metrics other than FPS, uncertainty is reported as ± two standard errors (SE), giving an approximate 95% confidence interval. Unless otherwise stated, we estimate SE using 1,000 nonparametric bootstrap resamples of the test image pairs. For HPatches, where accuracy is the proportion of homography estimates below a given corner-error threshold, we instead use the binomial standard error $\sqrt{p(1-p)/n}$, where $p$ is the measured accuracy and $n$ is the number of image pairs. We report only the aggregate scores returned by the RUBIK evaluation pipeline and therefore do not include uncertainty estimates for this benchmark.

### 3.6 Differences between paper & supplement and code

Table 1 summarizes the discrepancies we found between the paper and supplement on one side and the released code on the other, together with the choice adopted in our reproduction. We consider these choices the most faithful interpretation of the intended method.

## 4 Results

### 4.1 Main reproduction results

We begin by revisiting the central speed–accuracy experiment from the original paper. AUC@10° summarizes relative pose accuracy, with higher values indicating more accurate camera-pose estimates, while frames per second (FPS) measures inference throughput. This experiment examines whether sparse XFeat achieves the

Table 1: Summary of discrepancies between the XFeat paper & supplement and released code, and the choice adopted in our reproduction.

| Component | Paper & supplement | Released code | Our choice |
|---|---|---|---|
| Backbone depth | Six blocks, incl. early conv (supplement) | Omits one early conv | Follow supplement |
| Fusion block | Three basic layers | Two basic layers + $1 \times 1$ conv | Follow code |
| Keypoint head | Four conv layers, underspecified | Three basic layers + $1 \times 1$ classifier, no BN/ReLU on output | Follow code |
| Fine-matching loss $\mathcal{L}_{fine}$ | Cell classification, $8 \times 8$ grid | Confidence-weighted coordinate classification | Follow paper |
| Reliability loss $\mathcal{L}_{rel}$ | Single term, dual-softmax-derived confidence | Separate L1 loss to dual-softmax confidence | Follow paper |
| Keypoint loss $\mathcal{L}_{kp}$ | ALIKE distillation, weight $\delta$ | ALIKE distillation, equal-averaged | Follow paper |
| Loss weights $\alpha, \beta, \gamma, \delta$ | Symbolic, no values reported | Implicit equal weighting | Equal weighting |
| Non-parallel keypoint head (ablation) | "Conv block on encoder features," no detail | No such variant provided | Matched to reliability head design |

highest throughput among the learned methods and whether XFeat* improves pose accuracy at a moderate cost in speed.

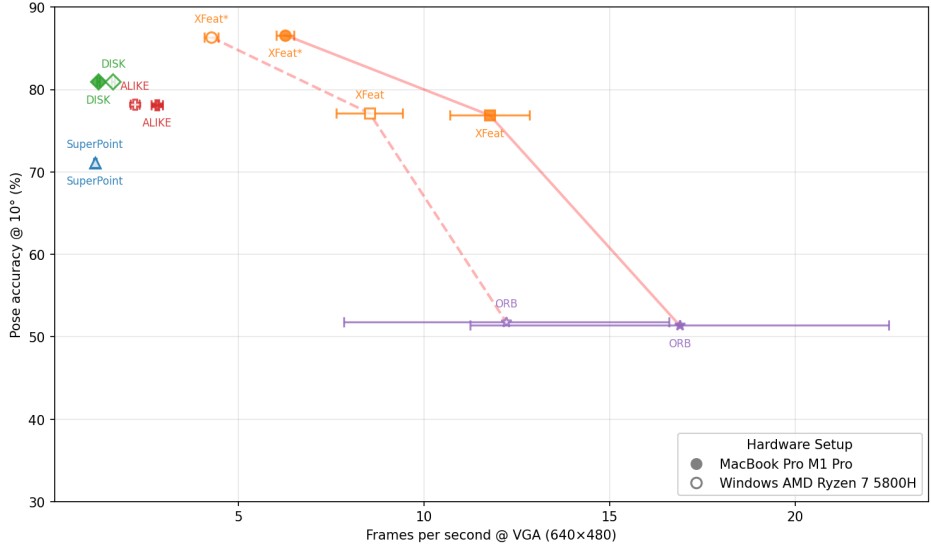

Figure 2: Speed–accuracy trade-off for relative pose estimation on MegaDepth-1500 under CPU inference. Irrespective of the platform, XFeat provides the highest throughput among the evaluated learned methods, while XFeat* improves pose accuracy at a lower but still comparatively high throughput.

Figure 2 (compare with Figure 1 in the original paper) and Table 2 recover the central speed–accuracy trade-off reported in the original paper. On Apple M1 Pro, XFeat reaches 11.8 FPS and an AUC@10° of 76.6, making it the fastest learned method while remaining competitive in pose accuracy. XFeat* increases AUC@10° to 86.0 at 6.3 FPS, trading roughly half the throughput for a substantial accuracy gain while remaining faster than the other learned baselines. ORB is the fastest method overall, but has considerably lower pose accuracy and more variable inference times. Although hardware and software differences prevent

Table 2: Speed and relative pose estimation accuracy on MegaDepth-1500 measured on Apple M1 Pro. The best results are in bold, and the second best are underlined.

| Method | AUC@10°↑ | FPS↑ |
|---|---|---|
| XFeat | 76.6 ± 2.20 | 11.8 ± 1.07 |
| XFeat* | **86.0** ± 1.78 | 6.3 ± 0.24 |
| ALIKE | 78.1 ± 2.14 | 2.8 ± 0.15 |
| DISK | 80.9 ± 2.02 | 1.2 ± 0.04 |
| ORB | 51.4 ± 2.58 | **16.9** ± 5.64 |
| SuperPoint | 71.1 ± 2.34 | 1.1 ± 0.02 |

direct comparison of absolute FPS values, the relative ordering of the methods and the overall speed–accuracy trade-off are preserved.

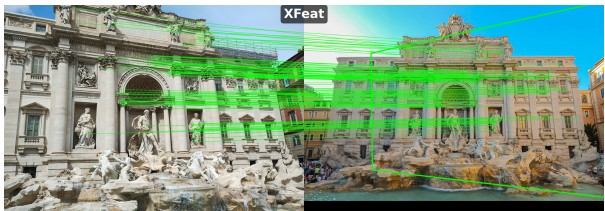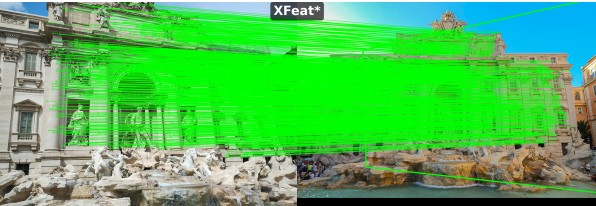

Figure 3: Qualitative comparison of correspondences produced by XFeat (left) and XFeat* (right) on the same image pair. XFeat and XFeat* were configured to return up to 1,012 and 16,000 correspondences, respectively; only matches retained as homography inliers by RANSAC are visualized. The semi-dense variant produces substantially denser correspondences with broader spatial coverage.

Figure 3 (compare with Figure 2 in the original paper) illustrates the difference between the two operating modes qualitatively. Sparse XFeat was configured to return up to 1,012 correspondences, whereas XFeat* used up to 16,000. XFeat* produces a substantially denser set with broader support. The quantitative results in Figure 2 show that these additional correspondences improve pose estimation at the cost of lower throughput. This is consistent with the original motivation for semi-dense refinement, which aims to provide more geometric constraints without the computational cost of fully dense matching.

Next, we test whether the MegaDepth-1500 and ScanNet-1500 results reported in the original paper can be recovered using both the authors' released checkpoint and our independently trained model. Small differences in the checkpoint re-evaluation may arise from the RANSAC-based evaluation and hardware differences. Acc@$\tau$° denotes the percentage of image pairs with a pose error below an angular threshold $\tau$, MIR denotes the mean inlier ratio, and #inliers is the average number of geometrically consistent matches.

Table 3: Relative camera pose estimation on MegaDepth-1500. We compare the original results with our checkpoint re-evaluation and independently trained reproduction. FPS for the re-evaluated and reproduced models is measured on an Apple M1 Pro.

| Method | AUC@5°↑ | AUC@10°↑ | AUC@20°↑ | Acc@10°↑ | MIR↑ | #inliers↑ | FPS↑ |
|---|---|---|---|---|---|---|---|
| XFeat Original | 42.6 | 56.4 | 67.7 | 74.9 | 0.55 | 892 | **27.1** ± 0.33 |
| XFeat Re-evaluated | 43.5 ± 1.96 | 57.1 ± 1.98 | 68.4 ± 1.88 | 75.1 ± 2.22 | 0.56 | 916 ± 28 | 12.9 ± 1.61 |
| XFeat Reproduced | 43.8 ± 1.96 | 57.9 ± 1.98 | 69.3 ± 1.86 | 76.6 ± 2.20 | 0.65 | 1090 ± 32 | 11.8 ± 1.07 |
| XFeat* Original | 50.2 | 65.4 | 77.1 | 85.1 | 0.74 | 1885 | 19.2 ± 1.12 |
| XFeat* Re-evaluated | **50.6** ± 1.82 | 65.7 ± 1.72 | 77.5 ± 1.50 | 85.3 ± 1.80 | 0.75 | 1905 ± 68 | 7.2 ± 0.38 |
| XFeat* Reproduced | 49.9 ± 1.88 | **65.8** ± 1.72 | **78.0** ± 1.44 | **86.0** ± 1.78 | **0.76** | **2159** ± 72 | 6.3 ± 0.24 |

Table 3 (compare with Table 1 in the original paper) shows that the reported pose-estimation accuracy is closely reproduced. For both XFeat and XFeat*, the re-evaluated checkpoint remains close to the original results, while the differences between the re-evaluated and reproduced models are within the reported uncertainty. The reproduced sparse model yields a higher mean inlier ratio and more inliers, although this does not translate into a measurable improvement in pose accuracy. For XFeat*, the reproduced and re-evaluated models are nearly identical across the main accuracy metrics. The largest discrepancy is in FPS, although the measurements were obtained on different processors.

Table 4: Relative camera pose estimation on ScanNet-1500. We compare the original results with our checkpoint re-evaluation and independently trained reproduction.

| Method | AUC@5°↑ | AUC@10°↑ | AUC@20°↑ |
|---|---|---|---|
| XFeat Original | 16.7 | 32.6 | 47.8 |
| XFeat Re-evaluated | $15.2 \pm 1.28$ | $30.3 \pm 1.68$ | $45.5 \pm 1.90$ |
| XFeat Reproduced | $16.5 \pm 1.32$ | $32.0 \pm 1.66$ | $47.5 \pm 1.86$ |
| XFeat* Original | 18.4 | 34.7 | 50.3 |
| XFeat* Re-evaluated | 17.8 $\pm 1.36$ | $33.2 \pm 1.74$ | $48.6 \pm 1.94$ |
| XFeat* Reproduced | **20.2** $\pm 1.38$ | **37.1** $\pm 1.70$ | **53.2** $\pm 1.82$ |

Table 4 (compare with Table 2 in the original paper) shows that the ScanNet-1500 results are also reproducible. For sparse XFeat, the reproduced model closely matches the original results and performs slightly better than the re-evaluated checkpoint across all three thresholds. The difference is more pronounced for XFeat*, where the reproduced model exceeds both the reported and re-evaluated results, increasing AUC@10° from 34.7 and 33.2 to 37.1. Together with the MegaDepth-1500 results in Table 3, these findings support the reproducibility of XFeat's main pose-estimation results across both outdoor and indoor scenes.

## 4.2 Ablations

The ablations test three implementation choices: whether keypoint detection should remain separate from descriptor extraction, whether XFeat benefits from its single skip-connection, and whether the loss formulation described in the paper differs meaningfully from the one implemented in the released code. The original paper does not release its ablation code and describes the coupled detector only as a convolutional block applied to encoder features. We therefore use the same head structure as the reliability branch, keeping the detector capacity comparable while varying whether keypoints are predicted from a separate image-encoding branch or from shared backbone features.

Table 5: Ablation results on MegaDepth-1500. We compare the default model with a coupled keypoint detector, three skip-connection variants, and the loss implemented in the released code.

| Method | AUC@5°↑ | AUC@10°↑ | AUC@20°↑ | Acc@10°↑ | MIR↑ | #inliers↑ |
|---|---|---|---|---|---|---|
| XFeat Default | $43.8 \pm 1.96$ | $57.9 \pm 1.98$ | $69.3 \pm 1.86$ | $76.6 \pm 2.20$ | 0.65 | $1090 \pm 32$ |
| XFeat Coupled Detector | $43.8 \pm 1.90$ | $57.5 \pm 1.94$ | $68.7 \pm 1.84$ | $75.9 \pm 2.20$ | 0.64 | $1105 \pm 34$ |
| XFeat No Skip | $42.3 \pm 1.94$ | $56.6 \pm 1.96$ | $68.2 \pm 1.88$ | $75.6 \pm 2.24$ | 0.64 | $1057 \pm 32$ |
| XFeat Input Skips | $43.8 \pm 2.00$ | $57.5 \pm 2.02$ | $69.0 \pm 1.88$ | $76.0 \pm 2.26$ | 0.65 | $1094 \pm 34$ |
| XFeat ResNet Skips | $44.7 \pm 1.96$ | $58.6 \pm 1.96$ | $69.9 \pm 1.86$ | $77.7 \pm 2.18$ | 0.64 | $1073 \pm 32$ |
| XFeat Code Loss | $45.5 \pm 1.96$ | $59.5 \pm 1.96$ | $70.4 \pm 1.88$ | $78.0 \pm 2.18$ | 0.59 | $996 \pm 15$ |
| XFeat* Default | $49.9 \pm 1.88$ | $65.8 \pm 1.72$ | $78.0 \pm 1.44$ | $86.0 \pm 1.78$ | 0.76 | **2159** $\pm 72$ |
| XFeat* Coupled Detector | $46.9 \pm 1.86$ | $62.6 \pm 1.76$ | $75.8 \pm 1.48$ | $84.3 \pm 1.84$ | 0.73 | $1974 \pm 66$ |
| XFeat* No Skip | **52.7** $\pm 1.82$ | **67.9** $\pm 1.66$ | **79.5** $\pm 1.38$ | **87.5** $\pm 1.66$ | 0.76 | $2116 \pm 70$ |
| XFeat* Input Skips | 51.5 $\pm 1.82$ | $67.0 \pm 1.66$ | $78.9 \pm 1.40$ | 87.3 $\pm 1.68$ | **0.79** | 2132 $\pm 72$ |
| XFeat* ResNet Skips | 51.5 $\pm 1.84$ | 67.2 $\pm 1.66$ | 79.0 $\pm 1.40$ | 87.3 $\pm 1.70$ | **0.79** | $2089 \pm 72$ |
| XFeat* Code Loss | $49.5 \pm 1.82$ | $65.5 \pm 1.66$ | $77.8 \pm 1.42$ | $86.5 \pm 1.78$ | 0.76 | $2036 \pm 34$ |

Table 6: Ablation results on ScanNet-1500 using the same variants as in Table 5.

| Method | AUC@5°↑ | AUC@10°↑ | AUC@20°↑ |
|---|---|---|---|
| XFeat Default | 16.5 ± 1.32 | 32.0 ± 1.66 | 47.5 ± 1.86 |
| XFeat Coupled Detector | 15.9 ± 1.30 | 30.9 ± 1.70 | 46.2 ± 1.94 |
| XFeat No Skip | 15.0 ± 1.30 | 30.4 ± 1.68 | 45.7 ± 1.88 |
| XFeat Input Skips | 15.3 ± 1.32 | 30.2 ± 1.68 | 45.7 ± 1.88 |
| XFeat ResNet Skips | 15.9 ± 1.26 | 30.8 ± 1.66 | 45.6 ± 1.82 |
| XFeat Code Loss | 15.2 ± 1.26 | 30.6 ± 1.66 | 46.1 ± 1.86 |
| XFeat* Default | 20.2 ± 1.38 | 37.1 ± 1.70 | 53.2 ± 1.82 |
| XFeat* Coupled Detector | 19.6 ± 1.44 | 36.7 ± 1.74 | 52.6 ± 1.84 |
| XFeat* No Skip | **20.3** ± 1.44 | 37.2 ± 1.78 | 53.1 ± 1.84 |
| XFeat* Input Skips | 19.7 ± 1.46 | 36.7 ± 1.76 | 53.1 ± 1.88 |
| XFeat* ResNet Skips | 19.8 ± 1.44 | 36.8 ± 1.76 | 53.2 ± 1.88 |
| XFeat* Code Loss | 20.2 ± 1.44 | **37.6** ± 1.78 | **53.8** ± 1.88 |

Tables 5 and 6 show that separating keypoint detection from descriptor extraction matters most for semi-dense matching on MegaDepth. In sparse mode, coupling the detector to the backbone changes pose accuracy only slightly on both datasets. For XFeat*, however, AUC@10° decreases from 65.8 to 62.6 on MegaDepth, while AUC@20° decreases from 78.0 to 75.8. The corresponding changes on ScanNet are small. These results support the use of a separate keypoint branch for semi-dense matching, but show that its benefit is less consistent across datasets than suggested by the original ablation.

The second conclusion relates to the presence of at least one skip-connection. On sparse XFeat, removing the original skip-connection reduces performance on both MegaDepth and ScanNet. The decrease is modest but consistent across metrics, somewhat supporting the supplement's claim that the original skip-connection was beneficial. However, the magnitude of the improvement is insufficient to serve as decisive evidence that skip-connections significantly enhance performance. The differences among the stronger skip-connection experiments are often comparable to the reported uncertainty. The small and dataset-dependent differences suggest that skip-connections do not play a major role in XFeat's performance. One possible explanation is that XFeat's shallow architecture enables efficient gradient flow without requiring skip-connections.

The remaining open question is whether the original paper's specific single-skip-connection design is optimal. Our evidence does not appear to support this claim. In semi-dense mode, the no-skip variant performs best on MegaDepth and is nearly identical to the default model on ScanNet, while the input and ResNet-style variants also remain close. We therefore find limited evidence that the specific single skip-connection is essential or preferable to the alternatives tested. Its effect appears to depend on the dataset and matching mode.

Regarding the loss formulation, the results do not show a consistent advantage for either the paper-defined or code-defined objective. On MegaDepth, the Code Loss variant improves all pose-accuracy metrics for sparse XFeat, but reduces MIR from 0.65 to 0.59 and the mean number of inliers from 1090 to 996. For XFeat*, the differences are small across all metrics. On ScanNet, Code Loss performs below the default for sparse XFeat, but slightly improves AUC@10° and AUC@20° for XFeat*. Since the direction of the effect changes across datasets and matching modes, we do not find evidence that either loss formulation is preferable.

Overall, the ablations provide stronger support for the separate keypoint branch than for the specific skip-connection design, while the loss discrepancy has no consistent effect on pose accuracy.

### 4.3 Reproduction of the original downstream evaluations

The original paper evaluates XFeat on homography estimation and visual localization. Together, they assess whether XFeat's accuracy–efficiency trade-off carries over to broader geometric vision applications.

For homography estimation on HPatches (Balntas et al., 2017), the original paper specifies MAGSAC++ (Barath et al., 2020), a RANSAC-based method that robustly estimates the homography

while reducing the influence of incorrect correspondences, but does not report its threshold. We tested several values and found that 1.5 produced the results closest to the original report, so we use it consistently across all HPatches experiments. Performance is reported as Mean Homography Accuracy (MHA) $@\tau$, the percentage of image pairs with a homography error below $\tau$ pixels (Zhao et al., 2022).

Table 7: Homography estimation on HPatches, reported separately for illumination and viewpoint changes. Baseline and original XFeat results are taken from the original paper, while our XFeat evaluations use a MAGSAC++ threshold of 1.5.

| Method | Illumination | | | Viewpoint | | |
| --- | --- | --- | --- | --- | --- | --- |
| | MHA@3↑ | MHA@5↑ | MHA@7↑ | MHA@3↑ | MHA@5↑ | MHA@7↑ |
| SiLK (Gleize et al., 2023) | 78.5 | 82.3 | 83.8 | 48.6 | 59.6 | 62.5 |
| SuperPoint (DeTone et al., 2018) | 94.6 | 98.5 | 98.8 | **71.1** | 79.6 | 83.9 |
| DISK (Tyszkiewicz et al., 2020) | 94.6 | **98.8** | **99.6** | 66.4 | 77.5 | 81.8 |
| ORB (Rublee et al., 2011) | 74.6 | 84.6 | 85.4 | 63.2 | 71.4 | 78.6 |
| ZippyPoint (Kanakis et al., 2023) | 94.2 | 96.9 | 98.5 | 66.1 | 76.8 | 80.7 |
| ALIKE (Zhao et al., 2022) | 94.6 | 98.5 | **99.6** | 68.2 | 77.5 | 81.4 |
| XFeat Original | **95.0** | 98.1 | 98.8 | 68.6 | 81.1 | 86.1 |
| XFeat Re-evaluated | 93.7 ± 2.88 | 97.5 ± 1.84 | 98.3 ± 1.56 | 68.8 ± 5.40 | 82.4 ± 4.44 | 85.8 ± 4.08 |
| XFeat Reproduced | 93.3 ± 2.96 | 97.9 ± 1.70 | 98.6 ± 1.40 | 66.4 ± 5.50 | 79.0 ± 4.76 | 84.1 ± 4.26 |
| XFeat* Re-evaluated | 92.6 ± 3.10 | 96.8 ± 2.08 | 98.6 ± 1.40 | 51.5 ± 5.82 | 74.9 ± 5.06 | 84.4 ± 4.24 |
| XFeat* Reproduced | 93.7 ± 2.88 | 97.5 ± 1.84 | 97.9 ± 1.70 | 65.8 ± 5.54 | 80.3 ± 4.64 | **88.8** ± 3.68 |

Table 7 (compare with Table 3 in the original paper) broadly reproduces XFeat's homography-estimation performance. For sparse XFeat, both the released checkpoint and our reproduced model remain close to the reported results across illumination and viewpoint changes. Performance is consistently higher on illumination sequences than on viewpoint sequences, where larger geometric changes make alignment more difficult.

The effect of semi-dense matching depends on the evaluation threshold. Our reproduced XFeat* performs similarly to sparse XFeat at MHA@3 and MHA@5, but achieves the highest MHA@7 on the viewpoint split. Since this pattern does not hold across all thresholds or for the re-evaluated checkpoint, we do not find a consistent advantage over sparse XFeat.

Table 8: Visual localization on Aachen Day–Night. Each value is the percentage of queries localized within the stated position and orientation error thresholds. We compare the original results with our checkpoint re-evaluation and independently trained reproduction.

| Method | Day | | | Night | | |
| --- | --- | --- | --- | --- | --- | --- |
| | (0.25m, 2°)↑ | (0.5m, 5°)↑ | (5m, 10°)↑ | (0.25m, 2°)↑ | (0.5m, 5°)↑ | (5m, 10°)↑ |
| SuperPoint (DeTone et al., 2018) | **87.4** | 93.2 | 97.0 | 77.6 | 85.7 | 95.9 |
| DISK (Tyszkiewicz et al., 2020) | 86.9 | **95.1** | **97.8** | **83.7** | **89.8** | **99.0** |
| ORB (Rublee et al., 2011) | 66.9 | 76.1 | 81.7 | 10.2 | 12.2 | 19.4 |
| ZippyPoint (Kanakis et al., 2023) | 80.7 | 88.6 | 93.7 | 61.2 | 70.4 | 79.6 |
| ALIKE (Zhao et al., 2022) | 85.7 | 92.4 | 96.7 | 81.6 | 88.8 | **99.0** |
| XFeat Original | 84.7 | 91.5 | 96.5 | 77.6 | **89.8** | 98.0 |
| XFeat Re-evaluated | 76.3 | 83.6 | 89.8 | 65.3 | 79.6 | 87.8 |
| XFeat Reproduced | 79.0 | 86.4 | 90.9 | 66.3 | 80.6 | 89.8 |

Aachen (Sattler et al., 2018) is the only original downstream evaluation for which we do not recover the reported results. We evaluate it with HLoc (Sarlin et al., 2019), a visual-localization pipeline that retrieves likely reference images, matches their local features, and estimates the query camera pose. We use the reported 1024-pixel resolution and retain the top 4096 keypoints. The original paper does not specify the remaining HLoc settings, such as image retrieval, reference-map construction, and geometric verification. Table 8 shows that both the released checkpoint and our reproduced model fall below the published values at every threshold, although our reproduction performs slightly better under the same setup. This suggests that the gap comes from differences in the HLoc configuration or other missing evaluation details rather

than from the reproduced model itself. We therefore consider the Aachen results only partially reproducible. XFeat remains effective for visual localization, but at a lower accuracy than reported in the original paper.

## 4.4 Out-of-distribution generalization

We next broaden the evaluation beyond the evaluations conducted in the original paper. We begin with RUBIK, a standardized benchmark that tests driving scenes under controlled changes in overlap, scale, and viewpoint, and FIRE, which evaluates retinal image registration under a much larger domain shift. RUBIK tests whether XFeat remains robust when scene overlap, scale, and viewpoint become difficult, while FIRE tests whether features learned from natural images transfer to a different visual domain.

In the RUBIK (Loiseau & Bourmaud, 2025) benchmark, an image pair is considered successfully matched when the estimated pose has a rotation error below 5° and a translation error below 2 meters. Table 9 compares the results reported in the RUBIK paper with our re-evaluation of the released XFeat checkpoint and our independently trained reproduction.

Table 9: Relative pose estimation on RUBIK. We compare the results reported in the RUBIK paper with our checkpoint re-evaluation and independently trained reproduction.

| Method | Success (%)↑ | Time (ms)↓ |
|---|---|---|
| ALIKED (Zhao et al., 2023)+ LightGlue (Lindenberger et al., 2023) | **36.8** | 45 |
| DISK (Tyszkiewicz et al., 2020) + LightGlue | 35.9 | 69 |
| SuperPoint (DeTone et al., 2018) + LightGlue | 35.7 | 43 |
| SIFT (Lowe, 2004) + LightGlue | 33.1 | 194 |
| DeDoDe v2 (Edstedt et al., 2024a) | 30.4 | 282 |
| XFeat | 14.2 | 54 |
| XFeat* | 15.1 | 82 |
| XFeat + LighterGlue | 30.1 | 43 |
| XFeat Re-evaluated | 14.5 | 38 |
| XFeat Reproduced | 15.2 | **37** |
| XFeat* Re-evaluated | 14.9 | 73 |
| XFeat* Reproduced | 16.0 | 79 |

On RUBIK, both XFeat variants generalize less effectively than the stronger baselines. Sparse XFeat reaches success rates of 14.5% for the released checkpoint and 15.2% for our reproduction, while XFeat* reaches 14.9% and 16.0%, respectively. These values closely match those reported in the RUBIK paper (Loiseau & Bourmaud, 2025). However, they remain well below methods paired with LightGlue, which achieve success rates of approximately 30-37%. This suggests that standalone XFeat is less robust to the large changes in overlap, scale, and viewpoint that are present in RUBIK.

FIRE (Hernández-Matas et al., 2017) introduces a larger domain shift by evaluating XFeat on retinal images. The task is to align images of the same eye so that blood vessels and other anatomical structures coincide. The dataset is divided into easy, moderate, and hard pairs based on image overlap and anatomical change. Following the SuperRetina (Liu et al., 2022a) protocol, we exclude `P37_1_2` and evaluate the remaining 133 pairs.

Each image is resized so that its longest side is 1536 pixels, preserving fine vascular detail and leveraging XFeat's efficiency with high-resolution images. We estimate a homography from the XFeat correspondences using RANSAC with a 40-pixel threshold in the original image coordinates. Registration error is computed at the annotated landmarks, and AUC is reported as the average success rate over mean landmark-error thresholds from 1 to 25 pixels.

Table 10 shows that XFeat transfers reasonably well to retinal registration when the image pairs retain sufficient overlap. Our reproduced sparse model reaches an AUC of 94.99% on the easy subset and 74.86% on the moderate subset, remaining close to the specialized and large matching baselines. Performance drops

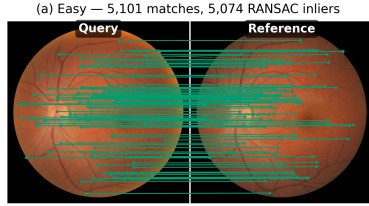 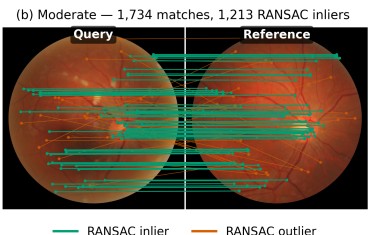 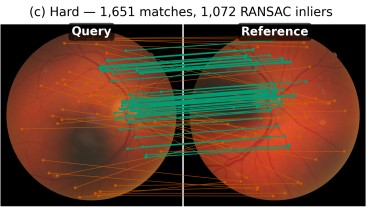

Figure 4: XFeat correspondences on easy, moderate, and hard FIRE image pairs. Green lines indicate matches retained as RANSAC inliers, while red lines indicate rejected matches. The easy examples yield substantially more geometrically consistent matches, whereas the moderate and hard examples yield fewer correspondences amid larger anatomical changes.

Table 10: Retinal image registration on FIRE. AUC is reported separately for the hard, moderate, and easy subsets.

| Method | AUC-Hard (%)↑ | AUC-Medium (%)↑ | AUC-Easy (%)↑ |
|---|---|---|---|
| SuperRetina (Liu et al., 2022a) | 54.20 | 78.30 | 94.00 |
| ASpanFormer (Chen et al., 2022) | **58.00** | 77.71 | 95.21 |
| ELoFTR (Wang et al., 2024) | 57.25 | 77.42 | 95.15 |
| GIM (Shen et al., 2024) | 56.75 | **79.14** | 95.38 |
| RoMa (Edstedt et al., 2024b) | 52.58 | 78.57 | 95.77 |
| MatchAnything$_{\text{ELoFTR}}$ (He et al., 2025) | 52.08 | 74.57 | 94.25 |
| MatchAnything$_{\text{RoMa}}$ (He et al., 2025) | 56.42 | 76.57 | **95.91** |
| XFeat Re-evaluated | $41.25 \pm 5.52$ | $73.71 \pm 8.68$ | $91.49 \pm 1.14$ |
| XFeat Reproduced | $49.92 \pm 3.94$ | $74.86 \pm 9.50$ | $94.99 \pm 0.86$ |
| XFeat* Re-evaluated | $30.33 \pm 6.34$ | $70.57 \pm 8.88$ | $86.82 \pm 0.86$ |
| XFeat* Reproduced | $42.17 \pm 5.02$ | $76.57 \pm 7.44$ | $89.58 \pm 0.86$ |

on the hard subset, with its AUC of 49.92% remaining below that of the strongest methods. This suggests that XFeat can transfer from natural images to retinal data, but struggles with pairs that have reduced overlap and larger anatomical variation.

Our reproduction outperforms the released checkpoint across all three subsets, with the largest gains on the hard pairs. XFeat* offers no clear overall benefit as it performs slightly better on the moderate subset, but worse than sparse XFeat on the easy and hard subsets. The examples in Figure 4 show the same general pattern, with many geometrically consistent matches in the easy pair and fewer correspondences as anatomical variation and overlap become more challenging. Overall, XFeat retains useful zero-shot matching ability on retinal images, but its generalization ability drops as sample difficulty increases.

### 4.5 Cross-modal transfer

We next consider the more challenging setting of cross-modal matching, where images of the same scene are captured using different sensing modalities. Unlike retinal experiments, where both images are obtained with the same sensor type, cross-modal pairs can differ substantially in appearance because each modality responds to different physical properties of the scene. We evaluate whether XFeat can establish reliable correspondences across thermal–visible aerial imagery and optical, infrared, and SAR remote-sensing images without additional training.

MTV (Liu et al., 2022b) evaluates matching between aerial thermal and visible images captured from different viewpoints. We evaluate the official test split, which contains 4,050 image pairs across five scenes. Each image is converted to grayscale and resized so that its longest side is 640 pixels. Following the MTV

protocol, matching precision is the proportion of correspondences consistent with the ground-truth epipolar geometry under a threshold of $10^{-4}$. Relative pose is estimated using the essential matrix and RANSAC, and performance is reported as AUC at pose-error thresholds of $5°$, $10°$, and $20°$. The published learning-based baselines were trained on MTV, whereas XFeat is evaluated in a zero-shot setting.

Table 11: Relative pose estimation on the official MTV thermal–visible test split. Published baseline results are taken from the MTV paper and were obtained after training on MTV, while all XFeat variants are evaluated without additional training. The best results are in bold, and the second best are underlined.

| Method | Precision (%)↑ | AUC@5° (%)↑ | AUC@10° (%)↑ | AUC@20° (%)↑ |
|---|---|---|---|---|
| LoFTR (Sun et al., 2021) | 66.81 | 6.9995 | 15.9352 | 27.4986 |
| QuadTreeAttention (Tang et al., 2022) | **78.97** | **10.2474** | **21.7289** | **34.5145** |
| D2-Net + NN (Dusmanu et al., 2019) | 16.17 | 0.0000 | 0.0000 | 0.0000 |
| SIFT + NN (Lowe, 2004) | 14.50 | 0.0000 | 0.0000 | 0.0000 |
| XFeat Re-evaluated | $7.62 \pm 0.35$ | $0.10 \pm 0.06$ | $0.46 \pm 0.15$ | $1.22 \pm 0.26$ |
| XFeat Reproduced | $9.07 \pm 0.44$ | $0.11 \pm 0.07$ | $0.48 \pm 0.15$ | $1.42 \pm 0.27$ |
| XFeat* Re-evaluated | $9.10 \pm 0.42$ | $0.11 \pm 0.07$ | $0.34 \pm 0.13$ | $0.90 \pm 0.23$ |
| XFeat* Reproduced | $10.76 \pm 0.50$ | $0.19 \pm 0.09$ | $0.58 \pm 0.17$ | $1.55 \pm 0.29$ |

Figure 5: Thermal–visible correspondences produced by XFeat on three MTV scenes. Green lines indicate matches that satisfy the ground-truth epipolar geometry, while red lines indicate geometrically inconsistent matches.

Table 11 shows that XFeat transfers poorly to thermal–visible matching. The reproduced sparse model achieves a matching precision of 9.07% and an AUC@20° of 1.42%, both substantially below those of LoFTR and QuadTreeAttention. The reproduced models perform slightly better than the released checkpoint, but all XFeat variants remain substantially below the MTV-trained baselines. Semi-dense matching provides only a small and inconsistent improvement. This suggests that increasing correspondence density alone is insufficient when cross-modal appearance differences prevent reliable descriptor matching.

Figure 5 shows that XFeat can occasionally recover correct matches when the scene geometry and thermal–visible appearance remain similar. On the race-track pair, XFeat recovers geometrically consistent correspondences, reaching a precision of 0.340. In the villa and country-road examples, larger geometric and thermal–visible appearance differences cause most matches to fail. Overall, XFeat can occasionally transfer when both the scene geometry and cross-modal appearance remain sufficiently similar, but it becomes unreliable under larger differences.

We further evaluate XFeat on three subsets of SRIF (Li et al., 2023), containing 200 optical–SAR, 200 optical–optical, and 200 optical–infrared image pairs. The optical–optical split provides a same-modality reference, while the other two splits test transfer across different sensor modalities. The image pairs include changes in scale, rotation, appearance, and spatial resolution, with a ground-truth affine transformation provided for each pair.

We follow the zero-shot evaluation pipeline of Corley et al. (2026). Each image pair is processed at its original resolution without tiling. Four intensity-normalization settings are considered to reduce appearance differences between modalities: no normalization, percentile rescaling, Z-score standardization, and contrast-

limited adaptive histogram equalization (CLAHE) (Zuiderveld, 1994). The XFeat correspondences are used to estimate an affine transformation with RANSAC using a 3-pixel threshold. The estimated transformation is compared with the ground truth over a regular grid of valid image points. MeanErr is the average displacement between their predicted and ground-truth locations, while S@5 and S@10 report the percentage of samples with errors below 5 and 10 pixels, respectively. Fail rate is the proportion of pairs for which no valid transformation is recovered. For each XFeat variant, we report the normalization with the lowest mean error over all 600 pairs and keep it fixed across the three subsets.

Table 12: Registration performance over all 600 SRIF image pairs. For each XFeat variant, we report the normalization with the lowest aggregate mean error. The best result is in bold, and the second best is underlined.

| Method | Norm | MeanErr (px)↓ | S@5 (%)↑ | S@10 (%)↑ | Fail rate (%)↓ |
|---|---|---|---|---|---|
| SuperPoint-LightGlue (Lindenberger et al., 2023) | CLAHE | 67.0 | 31.3 | 39.0 | 0.64 |
| RoMa (Edstedt et al., 2024b) | CLAHE | 64.9 | 36.0 | 40.4 | **0.00** |
| RoMa+LoFTR (Edstedt et al., 2024b) | CLAHE | 64.6 | 36.1 | 40.6 | **0.00** |
| RoMa+Tiny-RoMa (Edstedt et al., 2024b) | CLAHE | 63.9 | 35.3 | 40.0 | **0.00** |
| LoFTR (Sun et al., 2021) | CLAHE | 63.3 | 29.7 | 39.4 | 0.65 |
| LoFTR (Sun et al., 2021) | Z-Score | 63.1 | 30.6 | 41.7 | 0.64 |
| LoFTR (Sun et al., 2021) | Identity | 59.3 | 31.9 | 42.3 | 0.66 |
| MINIMA-RoMa Ren et al. (2025) | Identity | 48.2 | **42.0** | **45.8** | **0.00** |
| MINIMA-RoMa Ren et al. (2025) | CLAHE | 47.7 | 40.4 | 44.7 | **0.00** |
| MINIMA-RoMa Ren et al. (2025) | Z-Score | **47.0** | 41.7 | 45.7 | **0.00** |
| XFeat Re-evaluated | Z-Score | $90.1 \pm 6.19$ | $11.6 \pm 1.91$ | $17.4 \pm 2.36$ | **0.00** |
| XFeat Reproduced | Z-Score | $91.0 \pm 6.14$ | $12.3 \pm 1.93$ | $18.9 \pm 2.48$ | **0.00** |
| XFeat* Re-evaluated | CLAHE | $105.8 \pm 7.50$ | $10.0 \pm 1.84$ | $15.7 \pm 2.32$ | **0.00** |
| XFeat* Reproduced | Percentile | $101.4 \pm 8.30$ | $11.6 \pm 1.86$ | $17.8 \pm 2.30$ | **0.00** |

Table 12 shows that XFeat performs below the stronger matchers on SRIF. The reproduced sparse model reaches a mean error of 91.0 pixels, with 12.3 and 18.9% of the evaluated points falling within 5 and 10 pixels. Its success rates are slightly higher than those of the released checkpoint, although the mean errors are nearly the same.

Semi-dense matching does not improve the results. Relative to sparse XFeat, semi-dense matching increases the mean error by 17.4% for the released checkpoint and 11.4% for the reproduced model. Although all XFeat variants produce a valid affine transformation for every pair, the low success rates show that many of these estimates are inaccurate. The additional correspondences produced by XFeat* therefore do not resolve the large appearance differences in SRIF.

Table 13 highlights that XFeat performs best on the optical–optical subset, where the reproduced XFeat* model reaches an S@10 of 45.14%. However, XFeat's performance still degrades when moving to out-of-distribution aerial remote-sensing imagery. Performance drops further on optical–infrared pairs, with the reproduced sparse model reaching 13.39%, and is lowest on optical–SAR, where all variants remain below 2%. This suggests that XFeat generalizes poorly to out-of-distribution aerial imagery and becomes unreliable when the sensing modality changes, particularly for optical–SAR pairs, making it unsuitable for zero-shot multimodal remote-sensing registration without further adaptation.

Figure 6 shows that XFeat can recover accurate registrations for less challenging optical–optical pairs. However, performance remains inconsistent even within this subset, as shown by the large error in Figure 6 (f). The optical–SAR and optical–infrared examples contain occasional inliers, but these are generally insufficient to recover the correct global alignment. As on MTV, XFeat can match some pairs when the scene geometry and image appearance remain similar, but it becomes unreliable as these differences increase.

Table 13: Modality-specific SRIF registration performance of the re-evaluated and reproduced XFeat models. Each model uses the normalization selected from its aggregate 600-pair evaluation, which is kept fixed across all modality splits. The best result in each column is in bold, and the second best is underlined.

**(a) Optical–SAR**

| Matcher | Norm | MeanErr (px)↓ | S@5 (%)↑ | S@10 (%)↑ |
|---------|------|---------------|----------|-----------|
| XFeat Re-evaluated | Z-Score | **86.14 ± 4.60** | **0.47 ± 0.24** | **1.74 ± 0.80** |
| XFeat Reproduced | Z-Score | 89.12 ± 5.32 | 0.34 ± 0.11 | 1.55 ± 0.44 |
| XFeat* Re-evaluated | CLAHE | 98.00 ± 6.27 | 0.38 ± 0.20 | 1.48 ± 0.69 |
| XFeat* Reproduced | Percentile | 103.69 ± 8.01 | 0.33 ± 0.14 | 1.24 ± 0.40 |

**(b) Optical–Optical**

| Matcher | Norm | MeanErr (px)↓ | S@5 (%)↑ | S@10 (%)↑ |
|---------|------|---------------|----------|-----------|
| XFeat Re-evaluated | Z-Score | **112.88 ± 16.10** | 30.62 ± 5.22 | 41.82 ± 5.97 |
| XFeat Reproduced | Z-Score | 115.54 ± 15.82 | 30.21 ± 5.21 | 41.75 ± 6.19 |
| XFeat* Re-evaluated | CLAHE | 141.75 ± 20.82 | 26.18 ± 5.12 | 37.60 ± 6.09 |
| XFeat* Reproduced | Percentile | 120.39 ± 22.24 | **30.98 ± 5.19** | **45.14 ± 6.22** |

**(c) Optical–Infrared**

| Matcher | Norm | MeanErr (px)↓ | S@5 (%)↑ | S@10 (%)↑ |
|---------|------|---------------|----------|-----------|
| XFeat Re-evaluated | Z-Score | 71.32 ± 5.96 | 3.78 ± 1.68 | 8.70 ± 2.82 |
| XFeat Reproduced | Z-Score | **68.48 ± 6.51** | **6.28 ± 2.22** | **13.39 ± 3.61** |
| XFeat* Re-evaluated | CLAHE | 77.69 ± 7.69 | 3.53 ± 1.70 | 7.94 ± 2.74 |
| XFeat* Reproduced | Percentile | 80.26 ± 6.27 | 3.42 ± 1.72 | 6.91 ± 2.46 |

## 5 Discussion

### 5.1 Assessment of the original claims

We first assess the four claims defined in Section 2, considering both whether the reported results can be reproduced and whether the proposed architectural motivations are supported across datasets and tasks.

**Claim 1: XFeat provides a favorable accuracy–efficiency trade-off on resource-constrained hardware.** Our results support this claim. In the CPU benchmark, sparse XFeat is substantially faster than the other learned methods while maintaining competitive relative pose accuracy. XFeat* further improves pose accuracy by producing a larger set of refined correspondences, while retaining comparatively high throughput. Although our absolute FPS values are lower than those reported in the original paper, the relative ordering of the evaluated methods is preserved. The close agreement in pose accuracy also indicates that the reduction in inference speed is mostly due to the different hardware setup on which we evaluated the model. Overall, XFeat occupies a favorable efficiency niche among learned local feature methods, supporting the original paper's central claim.

**Claim 2: separating keypoint detection from descriptor extraction improves semi-dense matching.** Our results support this claim mainly for semi-dense matching. Coupling the detector to the descriptor backbone has little effect on sparse XFeat, but reduces XFeat* performance, particularly on MegaDepth. The separate branch therefore appears important when detected locations require refinement and when semi-dense matching must retain and refine a larger set of less certain candidate points, although its benefit is less consistent across datasets and matching modes than originally claimed.

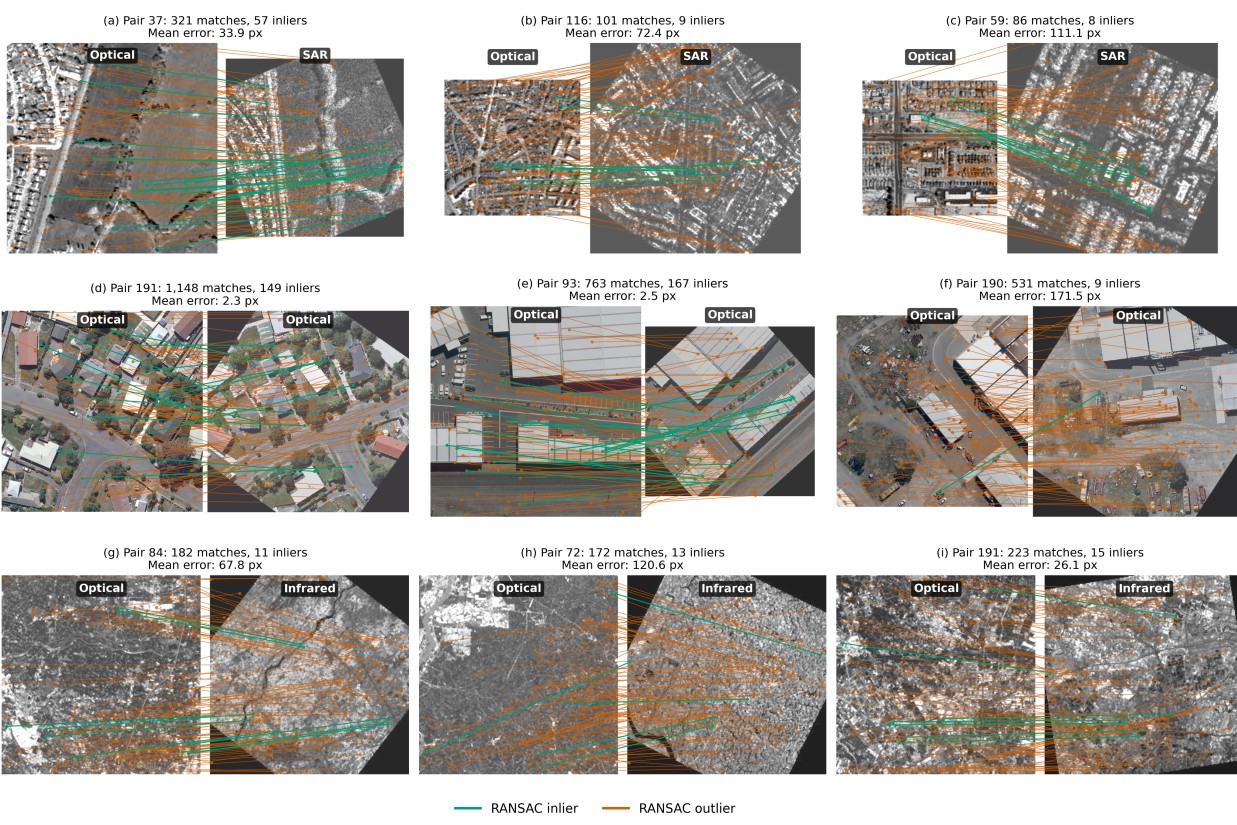

Figure 6: XFeat correspondences on optical–SAR, optical–optical, and optical–infrared SRIF image pairs. Green lines indicate RANSAC inliers and orange lines indicate rejected matches. Each panel reports the total number of correspondences, the number of inliers, and the mean registration error.

**Claim 3: the single skip-connection provides a measurable performance benefit and is preferable to alternative architectures.** Our results provide limited support for this claim. Removing the skip-connection reduces the accuracy of sparse XFeat on both MegaDepth and ScanNet, suggesting that passing early image information to the later backbone features can be beneficial. However, the differences are modest and are often comparable to the reported uncertainty. The semi-dense results are less consistent, as the model without a skip-connection performs best on MegaDepth and remains close to the default model on ScanNet.

We further tested whether the benefit is specific to the skip-connection design proposed in the original paper. Input projections added at several backbone stages, and ResNet-style residual connections generally match or slightly exceed the default architecture on MegaDepth, while producing small reductions on ScanNet. We therefore find no evidence that the original placement is consistently preferable to the alternatives we tested. One possible explanation is that the relatively shallow architecture does not benefit from the improved gradient flow introduced by skip-connections. Overall, the original skip-connection provides only a modest benefit in sparse matching and appears less influential than suggested in the original paper.

**Claim 4: XFeat provides competitive results on downstream geometric vision tasks.** This claim is supported for homography estimation and only partially supported for visual localization. On HPatches, both the released checkpoint and our independently trained model remain close to the reported results, confirming that XFeat transfers effectively from relative pose estimation to planar image alignment. The advantage of semi-dense matching is that it is less consistent, but the overall homography-estimation performance is reproducible.

In contrast, we are unable to reproduce the reported Aachen visual localization results. Both our reproduction and the released checkpoint perform worse than the original values at every threshold, suggesting that the gap is not primarily due to model training. It is more likely caused by unspecified parts of the localization pipeline, such as image retrieval, reference-map construction, or geometric verification. XFeat remains effective under our setup, but the published Aachen results are only partially reproducible without a more detailed description of the evaluation settings.

### 5.2  Out-of-distribution and cross-modal generalization

Our extended evaluation shows that XFeat does not generalize equally well across all domains. It remains effective on easier retinal image pairs, but performance drops as geometric and appearance differences increase, particularly on RUBIK, harder FIRE pairs, and cross-modal datasets. XFeat* provides little consistent gain in these settings, hinting that further correspondence refinement cannot compensate for the descriptors not being similar enough across modalities. Overall, XFeat handles moderate domain shifts reasonably well, but larger modality changes will likely require further adaptation or domain-specific fine-tuning.

### 5.3  What was easy

The model itself was relatively straightforward to rebuild and train. The released code and checkpoint served as a useful reference, while the supplementary material clarified most of the intended backbone and training choices. These resources made the core XFeat pipeline reproducible, although some architectural and evaluation details still required interpretation.

### 5.4  What was difficult

The hardest part was resolving ambiguities across the paper, supplement, and released code. Several architectural and training details required explicit interpretation, with the loss formulation being the clearest example because the written objective differs from the implemented training loop. Reproducing and extending the downstream evaluations was even more challenging. Aachen was computationally expensive, slow to debug, and sensitive to the localization configuration, while the incomplete HPatches setup required us to guess the evaluation hyperparameters. The additional out-of-distribution and cross-modal experiments also required separate preprocessing and evaluation pipelines for datasets with different image formats, annotations, and geometric protocols.

### 5.5  Communication with original authors

We did not contact the original authors; all implementation decisions were based solely on the paper, the supplement, the repository, and empirical validation using the released checkpoint.

## 6  Conclusion

This reproduction confirms XFeat's central message that a lightweight local feature model can achieve a strong accuracy-efficiency trade-off without specialized hardware-specific optimizations. We reimplemented the architecture and training objectives from the paper and supplementary material, re-evaluated the released checkpoint, and reproduced the main pose-estimation and downstream experiments. Our models closely match the released checkpoint on MegaDepth-1500 and ScanNet-1500, providing strong support for the method's core efficiency claim.

The additional experiments give a more nuanced view of the architectural choices. The separate keypoint branch is most beneficial for semi-dense matching, whereas the original skip-connection has a smaller and less consistent effect than the authors suggest. Homography estimation is reproduced closely, whereas Aachen visual localization depends strongly on unspecified evaluation details. Our out-of-distribution and cross-modal results further show that XFeat transfers reasonably under moderate domain shifts but degrades under severe appearance and modality changes. Overall, we conclude that XFeat remains a strong method for efficient natural-image matching.

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
