# OpenReview forum: "XFeat Revisited: Reproducibility and Evaluation of a Lightweight Image Matcher"
_TMLR — Under review for TMLR_

### Review · Reviewer_e6nT · 2026-06-24

**Summary Of Contributions:**

This paper proposes a reproducibility study of the method Xfeat (CVPR 2024). The paper evaluates running time performance, key designing choices (number of layers, dual-branch, …), and replicates results on downstream visual tasks. This evaluation is done with a comparison in three fronts: original paper numbers, replicating the experiments with provided model checkpoint, and with a custom reimplementation. The main conclusion is that the overall claims are properly verified except for the reproducibility of results with HLoc, which could have been due to omitted implementation details.

### Pros:

- The evaluation and reproducibility is convincing and covers most experiments of the main paper (ablations, downstream task).

- A more extensive ablation is performed considering core aspects of the method. These evaluations are also accompanied with statistical bounds on the replicated experiments.

- The motivation is clear, as well as the investigated main claims. The paper is also clear, well written and easy to follow.

### Cons:

- Although the provided experiments and evaluation is convincing to verify the claims and results of the original paper, it would be interesting to check and to perform comparisons on data affected by other conditions, that are present in real-world visual downstream tasks, such as when facing images presenting stronger view changes or that have been acquired in different domains (medical, remote sensing), e.g. RUBIK [R1] or sequences evaluated on MatchAnything [R2]. Another example is the suitability of this descriptor on data such as to perform localization from images to Gaussian Splatting representations such as GSplatLoc [R3].

  - [R1] “RUBIK: A Structured Benchmark for Image Matching across Geometric Challenges”, CVPR 2025. https://github.com/thibautloiseau/RUBIK

  - [R2] “MatchAnything: Universal Cross-Modality Image Matching with Large-Scale Pre-Training”, Arxiv 2025. https://zju3dv.github.io/MatchAnything/

  - [R3] “GSplatLoc: Grounding keypoint descriptors into 3D gaussian splatting for improved visual localization”. IROS 2025.

An evaluation of finetuning or retraining the method in other domains  (medical, remote sensing) would be interesting and alluring to the community.

- The title of the submission had better be changed to avoid confusion. It should be adapted to properly indicate that the focus is on reproducibility and evaluation, and I would recommend avoid using almost the same title of the original submission.

- The network architecture described in Fig. 1 could be improved. It is not clear where the outputs of the description and detection branches are.

**Audience:**

Yes

**Audience Explanation:**

There is a strong interest of the CV community for efficient visual matching approaches presenting a good tradeoff on performance to computational requirements. In that sense, this study of the reproducibility is informative since the recent trend and usability of Xfeat in several applications. Yet, it would have been informative to evaluate the method performance/comparison on other related tasks or datasets/images.

**Broader Impact Concerns:**

This point is not addressed and not applicable for this paper.

**Claims And Evidence:**

Yes

**Claims Explanation:**

The evaluation and experiments check the core results and claims of the original paper, both with the checkpoints and a custom from scratch re-implementation of the method. An interesting discussion is on the importance of the dual branch for the detection and description in the sparse and semi-dense modes. The overall evaluation is convincing.

**Requested Changes:**

Critical recommendations:

- This is a well described evaluation paper on the exact data/benchmarks adopted on the original paper. This is relevant but it would add interest to the paper if additional evaluations on other conditions/data was provided. What is the performance of the method when affected by stronger rotations, illumination changes or images from other domains (e.g., medical images, remote sensing, areal images, images to Gaussian Splating)? For instance an evaluation on the matching performance on datasets such as:

  - “RUBIK: A Structured Benchmark for Image Matching across Geometric Challenges”, CVPR 2025. https://github.com/thibautloiseau/RUBIK

  - “MatchAnything: Universal Cross-Modality Image Matching with Large-Scale Pre-Training”, Arxiv 2025. https://zju3dv.github.io/MatchAnything/

An evaluation of finetuning or retraining the method in other domains  (medical, remote sensing) would be interesting and alluring to the community.

- The title of the submission adds confusion. It should be adapted to properly indicate that this is a reproducibility and evaluation paper and not using the same title of the original paper. For instance something more like “Reproducibility and efficiency of Xfeat”?

---

> ### Author Response · Authors · 2026-07-08
> **Response to Reviewer e6nT**
>
> We thank the reviewer for the thorough and constructive feedback. We appreciate the positive assessment that our evaluation is convincing, covers the core claims of the original XFeat paper, includes additional ablations with statistical bounds, and is clearly motivated and written.
>
> We agree that evaluating XFeat under stronger geometric and domain shifts would further strengthen the paper. Our current evaluation already includes challenging settings with significant viewpoint and illumination changes, including downstream localization on Aachen Day-Night. However, we acknowledge that a more explicit evaluation on out-of-distribution and cross-domain data would be valuable for the wider computer vision community. In the revision, we are adding zero-shot evaluations on RUBIK, as well as additional cross-domain image-matching settings such as retinal images, aerial/thermal imagery, and SAR/remote-sensing data. We hope these experiments will help clarify how well XFeat’s efficiency and matching performance transfer beyond standard natural-image benchmarks.
>
> We also appreciate the reviewer’s suggestion regarding image-to-3D Gaussian Splatting localization. We note that GSplatLoc already directly explores this setting by using XFeat as a core descriptor component in a 3DGS-based localization pipeline, demonstrating XFeat’s strength in this direction. We believe that adding a dedicated 3DGS localization evaluation would largely amount to reproducing the GSplatLoc study, which is beyond the scope of our paper.
>
> Regarding fine-tuning or retraining in other domains, we agree that this is an interesting direction. However, our goal in this paper is to evaluate the reproducibility and practical behavior of XFeat as it is commonly used: as a lightweight, hardware-agnostic, general-purpose image matcher used out of the box. In that setting, zero-shot performance across challenging and out-of-distribution data is more directly aligned with the claims and expected use of this model.
>
> We  will change the title to avoid confusion with the original XFeat paper and make the reproducibility focus explicit, for example: “XFeat Revisited: Reproducibility and Evaluation of a Lightweight Image Matcher.”
>
> Finally, we acknowledge that Fig. 1 can be improved. We will update the architecture diagram to clearly indicate the outputs of the detection and description branches, including the keypoint scores, sparse descriptors, and semi-dense descriptor maps.
>
> We thank the reviewer again for the helpful suggestions. We believe these revisions will make the paper more useful to the wider computer vision community.

---

### Review · Reviewer_LRdQ · 2026-06-29

**Summary Of Contributions:**

This paper presents a reproducibility study of XFeat, a lightweight local feature detector and matcher for image correspondence tasks. The authors independently reimplement XFeat, re-evaluate the official released checkpoint, and conduct additional architectural ablations to assess the reproducibility of the original paper’s claims. The study also investigates inconsistencies between the original paper, supplementary material, and released code. The empirical evaluation covers the major benchmarks used in the original work, including MegaDepth, ScanNet, HPatches, and Aachen localization. The results show that the central accuracy-efficiency trade-off reported for XFeat is largely reproducible, particularly on MegaDepth and ScanNet, and that a reimplementation can closely match the performance of the official checkpoint. The ablation studies provide additional insight into the architecture, suggesting that the parallel keypoint branch is primarily beneficial for semi-dense matching (XFeat*) while the evidence supporting the original single skip-connection design is weaker than originally implied. The study also identifies a persistent reproducibility gap on Aachen localization that could not be fully resolved.

While I appreciate the effort put in by the authors in analyzing a published work, I strongly believe that the work is not aligned with the requirements of TMLR publication. Among other concerns listed under Additional Comments, there is no new algorithm proposed by the authors.

**Additional Comments:**

Weaknesses
•	The study appears to rely on single training runs, preventing estimation of training variability and limiting confidence in performance differences.
•	Reported uncertainty estimates appear to capture dataset-sampling variability rather than variability across independent training runs, potentially understating true uncertainty. I am also not sure whether the formula of binomial standard error as stated in the paper is correct for AUC. Also, why were uncertainty estimates computed analytically from dataset statistics rather than from repeated training runs? Can the authors provide confidence intervals or variance across independent trainings?
•	No statistical significance testing or multi-seed analysis is provided, making it difficult to determine whether several reported differences are meaningful.
•	Can the authors provide a dedicated ablation evaluating the impact of the training-loss discrepancies identified between the paper, supplement, and released code?
•	Can the authors provide a concise summary table listing all discrepancies found between the original paper, supplement, and code, along with the exact choices adopted in the reproduction?
•	Some conclusions are drawn from very small metric differences that may fall within expected variability.
•	The non-parallel keypoint ablation was modified relative to the original design, which weakens the extent to which conclusions can be attributed directly to the original architectural claim.
•	Important discrepancies in training losses between the original sources and released code are discussed but not directly evaluated through dedicated loss-function ablations.
•	HPatches results rely on a selected MAGSAC++ threshold, introducing potential evaluation-specific effects and raising questions about fairness of comparisons if competing methods were not similarly retuned.
•	The Aachen localization reproducibility gap remains unresolved; the proposed explanation involving underspecified HLoc details is plausible but not conclusively demonstrated.
•	Runtime comparisons are conducted on different hardware than the original work, which complicates direct interpretation of speed comparisons.
•	The paper could better situate itself within the broader machine learning reproducibility and replication literature.

**Audience:**

No

**Audience Explanation:**

The paper is simply a report on XFeat, a prior work published in CVPR. It is simply like a project report highlighting some potential issues with XFeat. However, there is no new algorithm that can make the paper strong enough to be interesting for TMLR audience.

**Claims And Evidence:**

No

**Claims Explanation:**

Check Weaknesses

**Requested Changes:**

I do not think the paper is suitable for publication without a new algorithm. Nevertheless, the authors can address the additional issues listed under Additional Comments.

---

> ### Author Response · Authors · 2026-07-08
> **Response to Reviewer LRdQ**
>
> We thank the reviewer for the detailed and constructive feedback and for engaging closely with the methodological details of our evaluation. We would like to first address the concern that the paper does not propose a new algorithm. We would like to respectfully note that reproducibility studies are explicitly within TMLR's scope, as reflected in the ML Reproducibility Challenge (https://reproml.org/) and previously published reproducibility work. Given the recent trend and usability of XFeat in several applications, we believe a study of its reproducibility is valuable for the community, separate from novel algorithmic contributions, and we hope the reviewer will consider the paper's value in that light.
>
> Regarding the reliance on single training runs, we agree that repeated training runs would enable estimation of training variability and increase confidence in performance differences. However, we note that the time required for enough training runs to get a sufficient sample size for quantifying uncertainty would be prohibitive (~30 days). Additionally, the original XFeat paper does not report using multiple seeds or repeated training runs either. Our reevaluations also indicate that training variability is not overly large, since we obtain similar results in almost all experiments.
>
> We agree with the reviewer regarding the binomial standard error formula for AUC and will revise our uncertainty reporting for pose-estimation metrics to use bootstrap standard errors (1000 resamples) instead of analytical formulas.
>
> Regarding statistical significance testing, we note that direct significance testing against a prior paper's results is uncommon practice in the reproducibility literature more broadly. Direct statistical comparison with results from the original paper would also be difficult because they are on the same datasets, so errors are correlated.
>
> We agree with the reviewer that a dedicated ablation evaluating the impact of the training loss discrepancies between the paper, supplement, and released code would be informative. We will add this ablation and state the choices and discrepancies more explicitly with a concise summary table.
>
> Regarding the non-parallel keypoint ablation, the original paper only specifies "a convolutional block on top of the encoder features" with no further detail, so an exact reproduction isn't possible. We gave this block the same architecture as the reliability head to keep the comparison as fair as possible, and will state this choice explicitly in the text.
>
> Regarding the HPatches MAGSAC++ threshold, we selected 1.5 based on what performed best for our XFeat variants, since the original paper doesn't report its threshold. The baseline numbers in Table 6 are taken directly from the original paper and were not retuned under our threshold; we will flag this explicitly as a limitation.
>
> Regarding the Aachen localization gap, the gap appears even with the original authors' own checkpoint, pointing to unspecified details in the HLoc pipeline rather than the model itself. We do not have a further resolution and will state this as an open problem.
>
> We agree with the reviewer that runtime comparisons on different hardware complicate direct interpretation of speed comparisons, but note that relative rankings between XFeat and baselines were consistent across hardware. Unfortunely, we do not have access to the same type of hardware used in the original study and we used the resourced available to us at the time.
>
> We thank the reviewer again for this thorough and technically substantive review.

---

### Review · Reviewer_rawQ · 2026-07-02

**Summary Of Contributions:**

This paper is a reproducibility study of XFeat, a lightweight local feature extractor and matcher. The study makes three key contributions:

Re-implementation: The authors provide a re-implementation of the XFeat architecture guided by the original paper and supplementary material, only referencing the official code when implementation-critical details were missing.

Re-evaluation: They re-evaluate the authors' released model across all datasets used in the original report and reproduce downstream experiments to verify whether XFeat's efficiency claims hold in practical real-world tasks.

Architectural Ablations: They conduct extensive ablations on:

Skip connections – finding that the motivation for a single skip connection is less conclusive than originally claimed.
Detector coupling – finding that the parallel keypoint branch is important for semi-dense matching, but its benefit is less pronounced than originally stated.
Overall, the study confirms XFeat's core efficiency claim — that a lightweight CNN can achieve a strong accuracy–efficiency trade-off without specialized hardware — while offering a more nuanced view of several architectural design choices.

Overall I support such papers that focus on evaluating State of the Art.

**Additional Comments:**

you can make it a worthy paper for TMLR if you modify your presentation to be more reader friendly. Make it a standalone paper.

**Audience:**

Yes

**Audience Explanation:**

While this is a Yes\No questions I think that a very narrow scope of people would be interested. Most of them need to have prior knowledge with the original Xfeat paper.
I personally dont have such knowledge, and this paper is not very accesible to me unless I dig in this knowledge. the overall application of such algorithms is of course of interest to the TMLR community.

**Claims And Evidence:**

Yes

**Claims Explanation:**

I think the paper confirms emprircally many of the claimed contributions of the original XFEAT paper.

**Requested Changes:**

You need to make the paper more accesible to the wider audiance:
1.  explain the problem,
2. explain some background on Xfeat,
3. motivate the structure  of xfeat
4. Stop using terminology and jargon that are mostly specific to Xfeat, unless you explain them

do that on the expense of over detailed techincal aspects that can be put in supplamentary material. Otherwise I dont think this paper is an easy read for TMLR audiance

---

> ### Author Response · Authors · 2026-07-08
> **Response to Reviewer rawQ**
>
> We thank the reviewer for the thoughtful and constructive feedback. We appreciate the positive assessment that the paper empirically verifies many of the core claims of the original XFeat work, and we are encouraged that the reviewer supports papers focused on evaluating state-of-the-art methods.
>
> We agree with the reviewer that the paper should be more accessible to a broader TMLR audience and should not assume detailed prior knowledge of the original XFeat paper. In the revision, we will make the paper more self-contained by expanding the introduction and background sections. We will more clearly explain the general problem of local feature extraction and image matching, why lightweight matching methods are important, and what practical role XFeat plays in this area.
>
> We will also revise the XFeat overview to motivate the architecture at a more abstract level before discussing implementation details. We will also reduce unexplained XFeat-specific terminology and make the main takeaways clearer.
>
> We believe these changes will make the paper read more like a standalone evaluation study and make it more useful to the broader audience.